# Prevalence of depression, anxiety and stress and their risk and protective factors among secondary students in Rwanda during the first wave of COVID-19 pandemic

Marie Bienvenue Mukantwali[1], Japhet Niyonsenga[1,2]*, Liliane Uwingeneye[3], Claudine Uwera Kanyamanza[1], Jean Mutabaruka[1]

1 Department of Clinical Psychology, College of Medicine and Health Sciences, University of Rwanda, Kigali, Rwanda, 2 Meantal Health and Behaviour Research Group, College of Medicine and Health Sciences, University of Rwanda, Kigali, Rwanda, 3 Department of Business Administration, College of Business and Economics, University of Rwanda, Kigali, Rwanda

* niyonsengajaphet74@gmail.com

## Abstract

### Introduction

Compelling evidence shows that the COVID-19 pandemic has detrimental effects on the mental health of university students. However, little is known about the psychological distress experienced by students from high schools during the pandemic. This study, therefore, sought to examine the prevalence of depression, anxiety and stress and their associated factors among students from high schools in Rwanda.

### Methods and materials

A retrospective, cross-sectional study was conducted on 384 students randomly selected from high schools. Data were collected using standardized measures of mental disorders and their associated factors. Bivariate and multivariate analyses based on the odds ratio were used to indicate the associated factors of anxiety, depression, and stress.

### Results

The results indicated that slightly above half of the participants (51%, n = 195) had clinically significant symptoms of depression, 30.3% (n = 116) had stress and 67.3% (n = 259) had anxiety. Our analyses identified several key risk factors associated with increased odds of these mental disorders. These include exposure to domestic violence, COVID-19 symptoms like cough and myalgia, eating twice per day, having one of the three mental disorders, gender, with females showing higher susceptibility, and direct contact with the people who positively tested covid-19. Conversely, protective factors such as heightened awareness about Covid-19, positive mental health, social support, eating three times, belonging to the third Ubudehe category, and a high resilience emerged as significant elements mitigating the risks of these mental health challenges within our sample. Intriguingly, religious affiliation

**Data Availability Statement:** All data are fully available from Zenodo via the link: https://zenodo.org/uploads/10957670.

**Funding:** The author(s) received no specific funding for this work.

**Competing interests:** The authors have declared that no competing interests exist.

emerged as a notable factor, with students affiliated with the Witness of Jehovah and Adventist denominations exhibited lower risks for depression and anxiety.

## Conclusion

Our findings highlighted a high prevalence of depression, anxiety, and stress among students from secondary schools. Interestingly, this study also revealed the associated risk and protective factors of depression, anxiety, and stress in Rwandan students in high schools. Therefore, mental health interventions targeting the impact of COVID-19 on students, as young people are needed.

## Introduction

At the end of 2019, the COVID-19 pandemic, which remains a public health emergency [1], began in China and later spread to all countries of the world [2]. The rapid spread of the epidemic and requiring extensive prevention and medical intervention efforts, significantly aggravated the pandemic's impact on human health, particularly mental health. This pandemic has profoundly affected psychological resilience among individuals and their communities, leading to several mental health issues such as anxiety, depression, loneliness, stress and phobia [3–5]. Earlier research indicates that the pandemic and associated quarantine measures have been the major contributors to psychological problems, leading to trauma and sleeping disorders [6–8]. This suggests a need for reinforced preventive measures to promote mental health [6–8].

Worryingly, studies have shown that the impact of COVID-19 is particularly severe among young people, including students, making them a vulnerable group [9]. Adolescence, being a developmental stage highly sensitive to stress exposures and the need for social interaction, has raised concerns regarding the pandemic's effect on adolescent mental health [9]. Factors such as high unemployment rates and the closure of schools during the epidemic, have contributed to a mental health crisis, causing significant psychological morbidity among vulnerable individuals [10]. Academic stress, known to trigger psychological reactions such as anxiety, anger, depression, and confusion among middle school students, may have been exacerbated in the face of the COVID-19 pandemic [10,11].

The psychological well-being of the school students has been notably affected due to the of schools' closure and the educational disruptions caused by these sudden changes. More than one-fifth of students of secondary schools reported a significant increase in negative mental health outcomes during COVID-19. Furthermore, the pandemic has negatively affected school students' psychological well-being, with reported increases in anxiety, depression, and stress levels [10,12]. Several mental disorders, including anxiety, depressive disorders, trauma, panic, and extreme fear, have been documented among students, affecting their quality of life and school performance. Contributing factors include disruptions in the academic calendar and a reduced interaction between staff, students and their families [13].

College students are increasingly recognized as a vulnerable population, suffering from higher levels of anxiety, depression, substance abuse, and disordered eating compared to the general population [14]. Studies conducted in Uganda [15] and Ethiopia [16,17] have revealed significant mental health challenges among higher education students, with high levels of anxiety, obsessive-compulsive, panic attacks, stress and depression. The factors associated include age, year of education, place of residence, and COVID-19 prevention practices, among others

[18,19]. The pandemic's impact has been particularly severe in countries like South Africa and Kenya, with significant morbidity, mortality and mental health impacts [20–22].

Despite COVID-19 being a significant public health concern [23], and the efforts by governments to combat its spread and its effects [24], few studies have focused on its impacts on mental health in Rwanda, particularly among students. A recent study in Rwanda on knowledge, attitude and practice towards COVID-19 among HIV patients showed that Rwandans are informed and have good attitudes towards COVID-19 [25]. However, the ongoing cases of COVID-19 continue to burden national socio-economic development and health sectors [26]. Another study in Rwanda has documented the pandemic's impact on Rwandans' mental health [24], highlighting challenges faced by individuals with pre-existing mental health disorders and those newly affected by the pandemic [27].

This study aimed to determine the prevalence of anxiety, stress, and depression and their associated risk and protective factors during COVID-19 among secondary school students. Specifically, it sought (1) to determine the prevalence of anxiety and depression during the COVID-19 period among the students from high schools in the Eastern Province; and (2) to examine the risk and protective factors of depression, anxiety, and perceived stress among the students from high schools in the Eastern Province. While numerous articles have emerged addressing various treatment options and clinical outcomes of patients during these outbreaks, the mental health status of different population groups must not be overlooked.

## Methods and materials

### Study design and setting

This was a retrospective, institutional-based, cross-sectional study conducted between January 2nd and February 4th, 2022. It aimed to assess the prevalence and risk factors of the psychological problems related to the COVID-19 pandemic among high school students. The study covered students from six high schools in the Eastern Province of Rwanda, specifically selected randomly across different school settings (boarding and non-boarding). The schools included were G.S Saint Vincent de Paul, G.S Saint Aloys, G.S Bihinga, G.S Gabiro High School, New Life Christian Academy and Ecole Secondaire Kayonza.

The Eastern Province, noted for extensive geography and low population density, comprises seven districts—Bugesera, Gatsibo, Kayonza, Ngoma, Kirehe, Nyagatare and Rwamagana, with a total of 248 high schools. For this study, three districts (Gatsibo, Rwamagana and Kayonza) were randomly selected, and within each, two schools were chosen through a random sampling method.

### Participants

Eligible participants were all students actively enrolled before the covid-19 outbreak in Rwanda on March 14, 2020, and aged between 14–24 years. Students who were seriously ill during data collection phase were excluded to ensure the accuracy and integrity of the response. A proportional stratification approach was adopted to select a sample that accurately reflected the diversity of the student population in the chosen districts. Sample size was calculated with Cochran's formula (2004),(), using parameters: 50% proportion ($p = 0.5$), 95% confidence intervals ($z = 1.96$), 5% margin of error ($e = 0.05$) [28]. Given the historically low non-response rate (<1%) in similar Rwandan populations, an adjustment for non-response was deemed unnecessary. This decision was based on empirical evidence suggesting minimal impact of non-response bias in our context. Thus, the final sample size was set at 384 participants, considered adequate for robust statistical analysis.

## Data collection and procedure

To ensure comprehension and cultural relevance, the research instruments were translated into Kinyarwanda and back-translated to English. Following a pre-test on 5% of samples (20 participants) from non-selected schools in Gasabo district, minor adjustments were made to the survey tools based on feedback. Data collection was conducted through in-person interviews by principal investigator and three research assistants (clinical psychologists), with school matrons facilitating the approach to the research participants. The following study tools were used:

## Socio-demographic questionnaire

Researchers gathered socio-demographic data, including the age, sex of the student, marital status, daily diet, religion, perception of being a student, social category, income, study level, parental status, hygiene and sanitation practices, perception of COVID-19, health insurance status, financial support during the lockdown, educational background, access to community assets, social media usage, experiences of domestic violence, and family size.

## Measure of physical symptoms

To assess physical symptoms experienced during the past lockdown and amidst the COVID-19 pandemic, participants were queried about symptoms such as dizziness, fever, vomiting, chills, headache, myalgia, cough, breathing difficulties, coryza, and sore throat. To measure its level of affecting, they were asked to rate the impact of these symptoms on health status and disclose any history of chronic medical conditions. Inquiries about the health services utilized in the 35 days included hospital admissions, quarantine experiences, and COVID-19 testing. Participants were also asked about their direct or indirect contacts with a confirmed COVID-19 case, individuals suspected of having COVID-19 or contaminated materials.

Regarding COVID-19 knowledge, the research focused on the sources of COVID-19 information, awareness of the COVID-19, confidence in diagnosis it, and satisfaction with health information received. Additionally, the study collected data on precautionary measures against the pandemic, including habits like avoiding sharing eating utensils, covering their mouth when coughing and sneezing, handwashing practices, mask usage and the average number of hours spent at home during the lockdown.

## The Perceived Social Support scale (PSS)

Participants evaluated the impact of COVID-19 on their social and family support networks by addressing support from received friends (a) and family(b), the sharing feelings with family members (c) and others (d), and care for the feelings of family members (e), [29]. Responses were categorized as much decreased, decreased, unchanged/same as before, increased, and much increased. The sum scores from these five items formed the PSS (Cronbach's Alpha, $\alpha$ = 0.66) with a range of 5–25. A lower score indicates diminished social and family support during the lockdown [30].

## The Warwick-Edinburgh Mental Well-being Scale (WEMWBS)

The WEMWBS measures mental wellbeing across three domains: personal, interpersonal and capacity [31]. The cut-off for this psychometric measure is documented as 44. This instrument, having shown satisfactory reliability in previous studies, demonstrated good consistency in this study (Alpha of Cronbach, $\alpha$ = 0.86).

### The Connor-Davidson Resilience Scale (CD-RISC-10)

The CD-RISC-10, consisting of 10 items on a 5-point Likert scale, measures resilience levels. It portrays individual feelings of ability and power in the face of difficulties [32]. This scale was rated on a 5-point Likert scale "1 = not true at all, to 5 = generally true". A higher score signifies greater resilience. A high score signifies a greater level of resilience. Consistent with prior studies [33], this study found the scale to exhibit good internal consistency (α = 0.82).

### The Depression, Anxiety Stress Scale (DASS-21)

The DASS-21, with 21 items divided among depression, anxiety, and stress subscales, assesses these mental health aspects using a 4-point Likert scale "0 = Did not apply to me at all to 3 = Applied to me very much" [34,35]. Recent findings have adjusted the cutoff scores for depression to 10, for anxiety to 8 and for stress to 15 [36]. Applying predefined limits, the study categorized the levels of anxiety, depression and stress into mild, moderate, moderate, severe, and extremely severe. The Cronbach's alpha values for depression (α = 0.8), anxiety (α = 0.76) and stress (α = 0.78) were satisfactory, with the overall DASS-21 scale showing high reliability (α = 0.91).

### Data analysis

Descriptive statistics were conducted for socio-demographic characteristics, physical symptoms, health services utilization, contact history with others, knowledge about the COVID-19 pandemic and precautionary measures. Bivariate regression analysis (BRA) was conducted to examine the relationship between mental health outcomes (depression, anxiety, stress, and mental wellbeing) and the other variables investigated. All significant variables within BRA models were incorporated into multiple logistic regression models based on likelihoods (or odds ratio) to assess the probability of mental health outcomes. The results were presented as odd ratios (ORs), where an odd ratio larger than 1 indicates a positive association and an odd ratio less than 1 signifies a negative association, along with 95% confidence intervals. All tests were two-tailed, with p<0.05 and 95% for significance and confidence intervals. Statistical analyses were performed using the Statistical Package for the Social Sciences (SPSS) version 28.

### Ethical considerations

This study adhered to the Helsinki Declaration principles, securing the approval from the Institutional Review Board (nᵒ: 394/CMHS IRB/2021) of the College of Medicine and Health Sciences (CMHS), University of Rwanda. Permission was also obtained from the Ministry of Education and participating schools. Informed consent (or assent for minors) was obtained from all participants after a thorough explanation of the study's purpose, benefits, and procedures.

## Results

### Demographic characteristics

A total of 384 secondary school students participated in the study, including 53.1% females. The participants were aged from 10 to 25 years (MA = 21.46 years, SD = 2.50). Many of the respondents (n = 210, 54.7%) were from households belong to the third category of Ubudehe. Only 37.2% (n = 143) and 40.4% (n = 155) experienced direct and indirect contact with covid-19 infected individuals, respectively (**Table 1**).

**Table 1. Demographic characteristics of participants.**

| Characteristics | Total (N = 384) | |
|---|---|---|
| | Number | Percent |
| **Gender** | | |
| Male | 180 | 46.9 |
| Female | 204 | 53.1 |
| Mean age (SD); min ±max | 18.02 (2.61); 12±23 | |
| **Age category** | | |
| 10–14 years | 44 | 11.5 |
| 15–18 years | 168 | 43.8 |
| 19–25 years | 172 | 44.8 |
| **Grade or year of education** | | |
| Grade 7 | 43 | 11.2 |
| Grade 8 | 15 | 3.9 |
| Grade 9 | 92 | 24.0 |
| Grade 10 | 97 | 25.3 |
| Grade 11 | 48 | 12.5 |
| Grade 12 | 89 | 23.2 |
| **Religion** | | |
| Catholic | 129 | 33.6 |
| Muslim | 79 | 20.6 |
| Adventist | 45 | 11.7 |
| ADEPR/EAR | 108 | 28.1 |
| **Ubudehe** | | |
| Category I | 33 | 8.6 |
| Category II | 141 | 36.7 |
| Category III | 210 | 54.7 |
| **Family income** | | |
| Farming | 190 | 49.5 |
| Sale | 84 | 21.9 |
| Livestock | 22 | 5.7 |
| Salary | 76 | 19.8 |
| Dons | 12 | 3.1 |
| **Experience direct contact with someone who tested positive for COVID-19** | | |
| No | 241 | 62.8 |
| Yes | 143 | 37.2 |
| **Indirect contact with a confirmed case of COVID-19** | | |
| No | 229 | 59.6 |
| Yes | 155 | 40.4 |

## Prevalence of domestic violence, positive Covid-19 test, depression, anxiety, and stress

Our results revealed that 37% of the students (n = 143) were from families that experienced domestic violence during the pandemic, while only 14.6% of the respondents tested positive for COVID-19. Our findings also indicated that slightly more than half of the participants (51%, n = 195) exhibited clinically significant symptoms of depression, 30.3% (n = 116) experienced stress and 67.3% (n = 259) had clinically significant symptoms of anxiety.

## Risk and protective factors of anxiety, depression, and stress

As demonstrated in Table 2, both bivariate and multivariate regression analyses were performed to examine the risk and protective factors associated with symptoms of depression, anxiety, and stress symptoms among Rwandan secondary schools' students during the Covid-19 pandemic.

## Risk and protective factors of depression symptoms

Our Multivariate Regression Analysis (MRA) revealed significant findings. Students in their senior years, specifically the 11th (OR = 8.47, 95% CI = 1.95–36.86, p<0.004) and 12th grades (OR = 2.02, 95% CI = 1.16–3.52, p<0.013), demonstrated a notably higher propensity towards depression symptoms. Additional risk factors such as exposure to domestic violence (OR = 4.25, 95% CI = 1.55–11.68, p = 0.005), being a COVID-19 patient (OR = 4.44, 95% CI = 1.8–10.91, p<0.001), and experiencing negative mental well-being (OR = 8.84, 95% CI = 2.58–30.22, p<0.001) significantly elevated the odds. Conversely, the absence of clinically significant anxiety symptoms (OR = 0.2, 95%CI = 0.07–0.55, p = 0.002) and stress (OR = 0.01, 95%CI = 0.00–0.06, p<0.001), alongside not exhibiting COVID-19 symptoms such as cough (OR = 0.17, 95%CI = 0.04–0.69, p = 0.013) and coryza (OR = 0.2, 95%CI = 0.1–0.41, p = 0.002), correlated with a reduced likelihood of depression.

## Risk and protective factors of anxiety symptoms

Our analysis further identified factors influencing anxiety symptoms. Testing positive for COVID-19 (OR = 3.38; 95%CI = 1.34–8.51, p<0.01], experiencing domestic violence (OR = 2.36, 95%CI = 1.13–4.96, p = 0.022), and being in the 9th (OR = 4.25, 95%CI = 1.55–11.68, p = 0.005), 11th (OR = 2.02, 95%CI = 1.16–3.52, p = 0.013] and 12th grades (OR = 4.4, 95%CI = 3.47–5.55, p<0.001) significantly increased anxiety risk. Interestingly, both being satisfied (OR = 2.5; 95%CI = 1.6–4.01, p<0.001) and very satisfied with student life (OR = 2.68, 95%CI = 1.9–3.85, p<0.001), and as well as negative mental health (OR = 2.52, 95%CI = 1.17–5.42, p = 0.018), and living in larger families (5–8 members), (OR = 3.78; 95%CI = 1.29–11.1) were linked to heightened anxiety. Conversely, the absence of clinically significant stress (OR = 0.05, 95%CI = 0.01–0.25, p<0.001), depression (OR = 0.15, 95%CI = 0.07–0.31, p<0.001), and COVID-19 symptoms like dizziness (OR = 0.35, 95%CI = 0.17–0.73, p = 0.005), myalgia (OR = 0.11, 95%CI = 0.05–0.24, p<0.001), sore throat (OR = 0.1, 95%CI = 0.03–0.43, p<0.001), coryza (OR = 0.25, 95%CI = 0.1–0.62, p = 0.003) and fever (OR = 0.22, 95% CI = 0.12–0.43, p<0.001) were associated with a reduced likelihood of anxiety symptoms.

## Risk and protective factors of stress symptoms

Regarding stress symptoms, factors increasing risk included tested positive for Covid-19 [OR = 5.91, 95%CI = 1.14–30.71, p = 0.003) and direct contact with COVID-19 infected individuals [OR = 3.33; 95%CI = 1.21–8.46, p = 0.011]. Notably, being in the 12th grades (OR = 3.61, 95%CI = 1.69–7.71, p<0.001) or 9th grades (OR = 3.13, 95%CI = 1.01–9.75, p = 0.049) and negative mental health (OR = 1.63, 95%CI = 1.5–1.83, p<0.001) significantly increased stress risk. Gender also played a role, with female having slightly higher odds of stress (OR = 1.14, 95%CI = 0.99–1.31, p<0.001). Protective factors included regular meal consumption, with students eating three times per day showing significantly lower stress levels (OR = 0.01, 95% = CI = 0.00–0.17, p<0.001). These findings illuminate the intricate interplay of various factors in shaping stress outcomes among high school students.

**Table 2. Risk and protective factors of anxiety, depression and stress among high school students.**

| | Anxiety | | Depression | | Stress | |
|---|---|---|---|---|---|---|
| Variables | Bivariate | Multivariate | Bivariate | Multivariate | Bivariate | Multivariate |
| | OR (95%CI) | OR (95%CI) | OR (95%CI) | OR (95%CI) | OR (95%CI) | OR (95%CI) |
| **Sex** | | | | | | |
| Male | 1 | 1 | 1 | 1 | 1 | 1 |
| Female | 1.79 (1.43–2.24)** | 2.09(1.06 -4.12)* | 0.43(0.28–0.66)* | 0.26(0.1–0.65) | 1.85(1.54–2.22)** | 1.14(0.99–1.31)** |
| **Age** | | | | | | |
| 10–14 years | 1 | | 1 | | 1 | 1 |
| 15-18years | 1.68 (0.84–3.36) | | 1.71(0.88–3.32) | | 1.77(0.87–3.58) | 2.76(0.72–10.51) |
| 19-25years | 1.19 (0.77–1.82) | | 0.85(0.55–1.32) | | 0.39(0.22–0.71)** | 0.39(0.14–1.08)* |
| **Grade** | | | | | | |
| Grade 7 | 1 | 1 | 1 | 1 | 1 | 1 |
| Grade 8 | 1.71 (1.5–1.98)** | 1.77(1.37–2.29)* | 1.55(1.47–2)** | 1.91(1.3–2.8)** | 1.81(0.9 -4.11 ) | 2.8(0.73–10.6) |
| Grade 9 | 2.82 (1.11–7.17)* | 4.25(1.55–11.68)** | 2.69(1.11–6.56)* | 1.71(0.35–8.36) | 2.43(1–5.93) | 3.13(1.01–9.75)* |
| Grade 10 | 5.54 (1.17–26.37)* | 1.24(0.3–5.05)** | 1.44(0.42–4.91) | 3.23(0.25–41.23) | 1.9(0.69–5.23) | 2.76(0.72–10.51) |
| Grade 11 | 2.26 (1.40–3.63)** | 2.02(1.16–3.52)* | 1.65(1.03–2.63)* | 8.47(1.95–36.86)** | 0.56(0.31–1.03) | 2.54(0.08–76.1) |
| Grade 12 | 2.99 (1.50–5.95) ** | 4.4(3.47–5.55)** | 2(1.2–3.34)** | 2.02(1.16–3.52)** | 1.75(0.63–4.87)* | 3.61(1.69–7.71)** |
| **Daily meals** | | | | | | |
| Once | 1 | 1 | 1 | 1 | 1 | 1 |
| Twice | 4.52 (2.29–8.91) ** | 6.38(4.77–8.6)** | 3.33(1.69–6.56)** | 1.52(0.32–7.17) | 3.27(1.55–6.91)** | 0.9(0.05–17.19) |
| Thrice | 2.75 (1.65–4.59) ** | 5.17(2.22–12.03)** | 2.52(1.44–4.39)** | 3.13(0.97–10.07) | 0.98(0.5–1.93) | 0.01(0–0.17)** |
| **Religion** | | | | | | |
| Catholic | 1 | 1 | 1 | 1 | 1 | |
| Muslim | 2.69 (1.06–6.79)* | 0.75(0.22–2.62) ** | 4.5(1.27–15.93)* | 1.9(0.24–15.03) | 1.68(0.46–6.1) | |
| Adventist | 0.44 (0.16–1.23) | 0.07(0.02–0.32)** | 1.56(0.41–5.95) | 6.79(0.7–65.86) | 0.75(0.18–3.1) | |
| ADEPR/EAR | 6.56 (2.16–19.90) ** | 1.3(0.29–5.92) | 6.97(1.81–26.8)** | 1.83(0.19–17.77) | 1.9(0.47–7.74) | |
| Witness of Jehova | 6.56 (2.49–17.32) ** | 0.73(0.19–2.8) | 10.08(2.82–36)** | 9.99(1.07–93.08)* | 2.56(0.71–9.26) | |
| **Family size** | | | | | | |
| Less than 5 | 1 | 1 | 1 | 1 | 1 | 1 |
| 5–8 members | 1.78 (0.85–3.71) | 3.78(1.29–11.1)* | 0.79(0.42–1.49) | 0.48(0.13–1.74) | 0.48(0.23–0.98)* | 0.01(0–0.27)** |
| More than 8 members | 0.37 (0.21–0.64) ** | 0.6(0.27–1.36) | 0.45(0.26–0.77)** | 2.29(0.86–6.05) | 0.29(0.16–0.53)** | 0.02(0–0.22)** |
| **Satisfaction with student's life** | | | | | | |
| Unsatisfied | 1 | 1 | 1 | 1 | 1 | 1 |
| Satisfied | 1.49 (1.23–1.82) ** | 2.5(1.6–4.01) | 1.34(0.67–2.66) | 1.87(0.18–19.51) | 1.48(0.64–3.41) | 0.36(0.05–2.54) |
| Very satisfied | 3.39 (1.68–6.84) * | 2.68(1.9–3.85)** | 0.57(0.28–1.18) | 3.3(0.32–33.62) | 0.51(0.2–1.28) | 0.48(0.06–3.69) |
| **Ubudehe** | | | | | | |
| Category I | 1 | | 1 | 1 | 1 | 1 |
| Category II | 0.57 (0.27–1.19) | | 0.26(0.1–0.67)** | 0.39(0.2–0.77)** | 0.4(0.19–0.85)* | 1.52(0.25–9.36) |
| Category III | 0.84 (0.55–1.30) | | 0.78(0.51–1.21) | 0.41(0.18–0.91)* | 0.48(0.28–0.84) | 0.15(0.04–0.62)** |
| **Family income** | | | | | | |
| Farming | 1 | 1 | 1 | 1 | 1 | 1 |
| Sale | 0.08 (0.01–0.61)* | 0.04(0.01–0.37)** | 0.05(0.01–0.39)** | 0.04(0–1.55) | 0.19(0.06–0.64)** | 0.26(0.06–1.03) |
| Livestock | 0.42 (0.05–3.49) | 0.4(0.04–3.78) | 0.2(0.02–1.65)* | 0.09(0–3.66) | 0.47(0.14–1.61) | 0.51(0.12–2.07) |
| Salary | 0.03 (0.00–0.32)** | 0.02(0–0.25)** | 0.01(0–0.11)** | 0(0–0.21)** | 0(0 | 0.31(0.08–1.28) |
| Dons | 0.13 (0.02–1.08) | 0.1(0.01–0.91)* | 0.03(0–0.27)** | 0.01(0–0.49)* | 0.21(0.06–0.74)* | 0.64(0.35–1.17) |
| **Family status** | | | | | | |
| Live with both parents | 1 | 1 | 1 | | 1 | |
| Live with only mother | 0.11 (0.01–0.88)* | 0.01(0–0.44)* | 3.79(0.82–17.6) | | 0.28(0.01–7.03) | |

(*Continued*)

**Table 2.** (Continued)

| Variables | Anxiety | | Depression | | Stress | |
|---|---|---|---|---|---|---|
| | Bivariate | Multivariate | Bivariate | Multivariate | Bivariate | Multivariate |
| | OR (95%CI) | OR (95%CI) | OR (95%CI) | OR (95%CI) | OR (95%CI) | OR (95%CI) |
| Live with father | 0.15 (0.02–1.20) | 0.04(0–1.04) | 3.44(0.71–16.82) | | 3.87(0.84–17.76) | |
| **Kandagira Ukarabe** | | | | | | |
| No | 1 | | 1 | 1 | 1 | |
| Yes | 1.29 (0.86–1.93) | | 1.83(1.21–2.75)** | 4.3 (1.6 -10.7 ) | 0.87(0.53–1.44) | |
| **Health insurance** | | | | | | |
| No | 1 | 1 | 1 | 1 | 1 | |
| Yes | 0.27 (0.10–0.69)** | 0.07(0.02–0.25)** | 0.21(0.06–0.73)* | 0.03(0–0.36)** | 0.18(0.02–1.35) | |
| **Received supports** | | | | | | |
| No | 1 | | 1 | | 1 | |
| Yes | 0.85 (0.46–1.59) | | 0.77(0.4–1.46) | | 0.85(0.38–1.9) | |
| **Accessing social media** | | | | | | |
| No | 1 | | 1 | | 1 | |
| Yes | 0.95 (0.63–1.43) | | 0.82(0.54–1.23) | | 1.66(1–2.77) | |
| **Domestic violence** | | | | | | |
| No | 1 | 1 | 1 | 1 | 1 | |
| Yes | 6.62 (2.03–21.55)** | 2.36(1.13–4.96)* | 3.62(1.7–7.71)** | 4.25(1.55–11.68)** | 0.22(0.08–0.61) | |
| **Tested COVID-19 positive** | | | | | | |
| No | 1 | 1 | 1 | 1 | 1 | 1 |
| Yes | 6.61 (2.03–21.4) ** | 3.38(1.34–8.51)* | 1.99(1.16–3.3)** | 4.44 (1.8 -10.91 )** | 4.44(1.8–10.91)** | 5.91(1.14–30.71)** |
| **Dizziness** | | | | | | |
| Yes | 1 | 1 | 1 | 1 | 1 | 1 |
| No | 0.32 (0.20–0.53) ** | 0.35(0.17–0.73)** | 0.45(0.29–0.7)** | 0.48(0.08–2.81) | 0.56(0.33–0.94)* | 1.441(0.248–8.376) |
| **Fever** | | | | | | |
| Yes | 1 | 1 | 1 | 1 | 1 | 1 |
| No | 0.25 (0.16–0.40) ** | 0.22(0.12–0.43)* | 0.33(0.22–0.5)** | 0.37(0.08–1.65) | 0.4(0.24–0.67)** | 5.102(0.8–32.557) |
| **Cough** | | | | | | |
| Yes | 1 | 1 | 1 | 1 | 1 | 1 |
| No | 0.61 (0.38–0.99) ** | 1.11(0.54–2.27) | 0.33(0.2–0.53)** | 0.17(0.04–0.69)* | 0.32(0.19–0.54)** | 0.02(0.001–0.212)** |
| **Vomiting** | | | | | | |
| Yes | 1 | 1 | 1 | 1 | 1 | |
| No | 0.31 (0.19–0.52) ** | 0.86(0.34–1.9) | 0.24(0.15–0.38)** | 0.71(0.18–2.83) | 0.67(0.39–1.14) | |
| **Chills** | | | | | | |
| Yes | 1 | 1 | 1 | | 1 | |
| No | 0.60 (0.39–0.92)* | 0.81(0.37–1.74) | 0.8(0.52–1.24) | | 0.92(0.54–1.57) | |
| **Headache** | | | | | | |
| Yes | 1 | | 1 | | 1 | 1 |
| No | 1.01 (0.65–1.56) | | 0.78(0.5–1.23) | | 0.49(0.27–0.91)* | 0.12(0.02–0.69)* |
| **Myalgia** | | | | | | |
| Yes | 1 | 1 | 1 | 1 | 1 | 1 |
| No | 0.28 (0.16–0.50) ** | 0.11(0.05–0.24)** | 0.19(0.11–0.32)** | 0.56(0.07–4.28) | 0.42(0.24–0.73)** | 0.75(0.07–7.62) |
| **Breathing** | | | | | | |
| Yes | 1 | 1 | 1 | 1 | 1 | |
| No | 0.45 (0.25–0.79) ** | 0.48(0.19–1.24) | 0.42(0.24–0.71)** | 1.69(0.23–12.66) | 0.64(0.35–1.17) | |
| **Sore throat** | | | | | | |
| Yes | 1 | 1 | 1 | 1 | 1 | |

(*Continued*)

**Table 2.** (Continued)

| Variables | Anxiety | | Depression | | Stress | |
|---|---|---|---|---|---|---|
| | Bivariate | Multivariate | Bivariate | Multivariate | Bivariate | Multivariate |
| | OR (95%CI) | OR (95%CI) | OR (95%CI) | OR (95%CI) | OR (95%CI) | OR (95%CI) |
| No | 0.56 (0.32–0.97)* | 0.11(0.03–0.43)** | 0.37(0.22–0.64)** | 0.51(0.09–2.86) | 0.59(0.33–1.08) | |
| **Coryza** | | | | | | |
| Yes | 1 | 1 | 1 | 1 | 1 | 1 |
| No | 0.56 (0.31–1.02) | 0.25(0.1–0.62)** | 0.38(0.21–0.68)** | 0.2(0.1–0.41)** | 0.43(0.23–0.8)** | 0.69(0.06–8.17) |
| **Direct contact with a person with COVID-19** | | | | | | |
| No | 1 | 1 | 1 | | 1 | 1 |
| Yes | 2.36 (1.13–4.96)* | 2.3(1.11–4.8)* | 0.79(0.52–1.2) | | 1.63(1.18–2.25)** | 3.33(1.21–8.46)* |
| **Awareness on covid-19** | | | | | | |
| No | | | 1 | | 1 | |
| Yes | 0.22 (0.07–0.69)* | 0.25(0.1–0.62)** | 0.29(0.08–1.04) | | 0.63(0.36–1.1)** | |
| **Sufficient knowledge of COVID-19** | | | | | | |
| No | 1 | 1 | 1 | | 1 | 1 |
| Yes | 0.55 (0.36–0.84)** | 0.62(0.26–1.46) | 0.73(0.47–1.13) | | 0.17(0.08–0.32) ** | 0.46(0.07–3.15) |
| **Anxiety** | | | | | | |
| Yes | NA | NA | 1 | 1 | 1 | 1 |
| No | NA | NA | 0.07(0.04–0.12)** | 0.2(0.07–0.55)** | 0.04(0.01–0.12)** | 0.2(0.05–0.83)* |
| **Depression** | | | | | | |
| Yes | 1 | 1 | NA | NA | 1 | 1 |
| No | 0.07 (0.04–0.12)** | 0.15(0.07–0.31)** | NA | NA | 0.03(0.01–0.07) ** | 0.03(0.01–0.14)** |
| **Stress** | | | | | | |
| Yes | 1 | 1 | 1 | 1 | NA | NA |
| No | 0.04 (0.01–0.12)** | 0.05(0.01–0.25)** | 0.03(0.01–0.07)** | 0.01(0–0.06)** | NA | NA |
| **Resilience** | | | | | | |
| High resilience | 1 | 1 | 1 | 1 | 1 | 1 |
| Low resilience | 1.85 (1.54–2.22)** | 2.48(2.11–2.93)** | 1.61(0.93–2.75)** | 1.68(1.09–2.91)** | 1.42(1.01–1.98 )* | 3.56(2.74 -4.62 )** |
| **Mental wellbeing** | | | | | | |
| Positive | 1 | 1 | 1 | 1 | 1 | 1 |
| Negative | 2.14 (1.38–3.31)** | 2.52(1.17–5.42)* | 1.811.192.76** | 8.84(2.58–30.22)** | 1.421.21.7)* | 1.631.51.83)** |

*:p<0.05

**:p<0.01.

## Discussion

This study explored the prevalence of mental health outcomes (depression, anxiety and stress) and their associated risk and protective factors among the students from high schools in Rwanda during the Covid-19 pandemic. Our findings indicated that 51%, 30.3%, and 67.3% of our sample reported clinically high levels of depression, stress, and anxiety, respectively. Despite the scarcity studies conducted exclusively on secondary students, the prevalence of depression, stress and anxiety in our study is lower yet comparable to those reported in the general population of Bangladesh [37], college students in Ethiopia [19] and primary and secondary school students in the Gaza Strip in Palestine [12]during the lockdown.

According to Abir et al.[37], approximately 64%, 87%, and 61% of the respondents in Bangladesh had clinically significant symptoms of depression, anxiety, and stress, respectively. Similarly, 77.2%, 71.8% and 48.5% of Ethiopian college students reported high levels of depression, anxiety and stress during the lockdown, respectively [19]. Among residents of Tepi town

in Ethiopia, the prevalence of depression, anxiety, and stress symptoms was 37.7%, 39.0%, and 44.2%, respectively [38]. In the Gaza Strip, the majority of the students experienced anxiety (88.4%) and depression (72.1%), while more than one-third (35.7%) experienced stress [12]. However, the prevalence in the current study is significantly higher than that reported among undergraduate students in Saudi Arabia, where anxiety, depression and stress were reported at 22.2%, 25.4%, 17.9%, respectively [39]. This difference may be attributed to the varying education levels of the samples, retrospective nature of this study, and the multitude of risk factors investigated in this study.

As revealed in previous studies, our findings highlighted that negative mental well-being, being infected with COVID-19, direct contact with Covid-19 infected individuals [40], and experiencing symptoms of COVID-19 (sore throat, coryza, myalgia, fever and dizziness) [41], being in the 11[th] grades or 12[th] grades [42], exposure to domestic violence and being female [43–46] increased the odds to have depression [47], anxiety [47] and stress. Students in the 9[th] and 12[th] grades faced higher odds of experiencing these mental disorders, possibly due to the pressures of preparing for the national exam, and frequent exposure to social media platforms spreading rumours, and biased information about COVID-19. Additionally, the closure of schools during the COVID-19 pandemic likely exacerbated concerns over educational achievements [48]. Our study confirmed that female students were more prone to depression, anxiety and stress, which may be explained by differences in risk perception [43]. This finding aligns with previous research indicating that female participants are more susceptible to psychological problems during COVID-19 [44,45]. Experiencing COVID-19 symptoms [41] was significantly associated with increased odds of depression, anxiety and/or stress. The pandemic's impact on daily life, including restrictions on social interaction and financial constraints, likely contributed to psychological distress among students.

Interestingly, students who had one of the three mental disorders had greater odds to have another mental disorder, suggesting a comorbidity among these conditions [49]. Another significant finding was that the students from families in the second social category were more likely to suffer from these mental health issues, a result consistent with studies highlighting the impact of low-income on mental health during the pandemic [50,51]. This result could be attributed to the fact that low-income families have limited access to daily necessities and materials required for preventive procedures during the pandemic [50,52]. Contrary to previous studies [51,53], we found that students from larger families were less likely to experience mental disorders.

Consistent with previous research, our findings identified several protective factors against depression, anxiety, and stress among high school students in Rwanda during the COVID-19 pandemic. These include awareness of Covid-19 [54], social support [55,56], being in the third Ubudehe category, positive mental health [57] and high resilience [58–60]. Echoing Zhang et al.[58], resilience emerges as key mediator that, alongside positive emotions can significantly enhance their coping mechanisms and overall well-being under stress (i.e. the COVID-19 pandemic), [61]. Additionally, the strong negative correlation between social support and symptoms of mental distress reinforces the importance of a supportive network during crises, highlighting its role in mitigating the psychological impacts [55,56]. These insights emphasize the need for integrated strategies that prioritize mental health support, resilience-building, and the fortification of social support systems to combat the adverse mental health effects of public health emergencies effectively [62].

## Strengths and limitations

The study's strengths lie in its pioneering examination of the mental health outcomes among Rwandan secondary students during the COVID-19 pandemic, offering invaluable insights for

the policymakers and shedding light on the pandemic's impact on the youth. Furthermore, its retrospective nature provides historical context, enriching our understanding of the evolving mental health landscape amidst the crisis. However, alongside its contributions, the study faces limitations, notably its retrospective nature, which, while offering valuable historical insights, may be constrained by the accuracy of recall among participants. Additionally, the cross-sectional design poses challenges in establishing causal relationships between pandemic-related factors and observed mental health outcomes. Therefore, longitudinal studies and real-time data collection efforts are essential to accurately capture the pandemic's long-term effects and inform more effective interventions and support systems for youth mental health.

## Conclusion

This study underscores the high prevalence of depression, anxiety and stress among secondary school students during the COVID-19 pandemic. Identified risk factors include domestic violence, COVID-19 symptoms, co-morbidities among the mental health outcomes, gender and direct contact with infected individuals. Conversely, awareness of Covid-19, positive mental health, social support, belonging to the third Ubudehe category, and high resilience were protective. The findings call for targeted interventions by policymakers and decision-makers, emphasizing the need for widespread vaccination and the cultivation of resilience and positive mental health during crises.

## Acknowledgments

The authors would like to thank the study participants for their participation and for facilitating the fieldwork throughout the study time.

## Author Contributions

**Conceptualization:** Marie Bienvenue Mukantwali, Japhet Niyonsenga, Claudine Uwera Kanyamanza, Jean Mutabaruka.

**Data curation:** Marie Bienvenue Mukantwali, Japhet Niyonsenga.

**Formal analysis:** Japhet Niyonsenga, Liliane Uwingeneye, Jean Mutabaruka.

**Investigation:** Marie Bienvenue Mukantwali.

**Methodology:** Marie Bienvenue Mukantwali, Japhet Niyonsenga, Claudine Uwera Kanyamanza, Jean Mutabaruka.

**Project administration:** Liliane Uwingeneye.

**Supervision:** Claudine Uwera Kanyamanza, Jean Mutabaruka.

**Writing – original draft:** Japhet Niyonsenga.

**Writing – review & editing:** Marie Bienvenue Mukantwali, Japhet Niyonsenga, Liliane Uwingeneye, Claudine Uwera Kanyamanza, Jean Mutabaruka.

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
