## [Decision Letter · Decision Letter 0]

3 Jan 2024

PONE-D-23-41465The prevalence of depression, anxiety, and stress and their risks and protective factors among secondary students in Rwanda during the first wave of the COVID-19 pandemic.PLOS ONE

Dear Dr. Niyonsenga,

Thank you for submitting your manuscript to PLOS ONE. After careful consideration, we feel that it has merit but does not fully meet PLOS ONE’s publication criteria as it currently stands. Therefore, we invite you to submit a revised version of the manuscript that addresses the points raised during the review process.

We look forward to receiving your revised manuscript.

Kind regards,

Wudneh Simegn, MSc

Academic Editor

PLOS ONE

Journal Requirements:

3. In the online submission form, you indicated that [Insert text from online submission form here]. 

Reviewers' comments:

Reviewer's Responses to Questions

**Comments to the Author**

1. Is the manuscript technically sound, and do the data support the conclusions?

Reviewer #1: Yes

Reviewer #2: Yes

2. Has the statistical analysis been performed appropriately and rigorously? 

Reviewer #1: Yes

Reviewer #2: Yes

3. Have the authors made all data underlying the findings in their manuscript fully available?

Reviewer #1: Yes

Reviewer #2: No

4. Is the manuscript presented in an intelligible fashion and written in standard English?

Reviewer #1: Yes

Reviewer #2: Yes

5. Review Comments to the Author

Reviewer #1: Dear Authors,

With great interest, I reviewed your manuscript entitled “The prevalence of depression, anxiety, and stress and their risks and protective factors among secondary students in Rwanda during the first wave of the COVID-19 Pandemic.’’ I recommend that you address the following suggestions for further development. Minor comments

1. What is the rational for selecting equal number of participants (128 each) from three districts (disproportional recruitment)?

2. Line 152 “Qualified health experts from the schools selected for this study…”. Are there Health experts in primary school in Rwanda?, why?

3. Language editing by native speakers

Major comments

1. The first wave of COVID-19 occurred in late 2019 and early 2020, however this study was conducted three years later, in 2022. Participants may not remember the exact situation during the first wave after three years. This may have an impact on the study's generalizability.

2. Re design all tables and use appropriate font size for texts

3. Make the discussion more concise, keeping the main points

Reviewer #2: #PONE-D-23-41465

Title: The prevalence of depression, anxiety, and stress and their risks and protective factors among secondary students in Rwanda during the first wave of the COVID-19

Reviewer reports

Thank you for the opportunity to review the manuscript that reports “the prevalence of depression, anxiety, and stress and their risks and protective factors among secondary students in Rwanda during the first wave of the COVID-19”. I have included some points below that should be addressed before considering the paper for potential publication in the journal.

• Your study result showed that, common associated risk factors were domestic violence, the existence of some symptoms of COVID-19 such as cough and myalgia, co-morbidities between depression, anxiety and stress, religion, being female and direct contact with the people who positively tested covid-19. How religion is associated with anxiety, stress, and depression? Is it logical?

• Line 9, at the affiliation of authors, you used code 3, as “ 3Department of Business Administration, College of Business and Economics, University of 10 Rwanda, Kigali, Rwanda”, however no author with such code on list of authors. The contribution of the author from the Department of Business Administration, College of Business and Economics, appears to be outside their discipline. It may be necessary to clarify the relevance of their contribution to the research.

• Line 137, When substituting the values for p and 1-p in the sample size determination formula on line 137, ensure that the correct p-value is used. Double-check and correct any missing or incorrect values to ensure accuracy in the calculation.

• While determining sample size, why non-response rate is not considered? Non-response rate is often considered when determining sample size, as it can impact the representativeness of the sample. Ignoring non-response rate could lead to biased results and affect the generalizability of the findings.

Methods

• This institutional-based, cross-sectional study was conducted from 2nd January to 4th April 2022….. The duration of the study (January to April 2022) may impact the response, as the prevalence and severity of symptoms could have changed since 2019-2020. How do you solve this problem, since the duration may affect the response?

• Line 192, …..we found that the consistency was also good (Alpha of Cronbach, α=0.86) (Error! Reference source not found.). check table citation problem and correct it.

• It seems that prior study during pandemic https://doi.org/10.1007/s40615-021-01195-1 not discussed. Check and discuss with your study.

• Please add the direction for the future work based on your work and prior studies.

• Line 389, <<this conducted="" declaration="" following="" helsinki="" study="" was="">> I don’t think that putting reference for this declaration is important. I suggest to remove reference.

It's important to thoroughly check the document for language, typos, grammar, and accurate interpretation of findings. This ensures the overall quality and credibility of the document.</this>

6. PLOS authors have the option to publish the peer review history of their article (what does this mean?). If published, this will include your full peer review and any attached files.

Reviewer #1: No

Reviewer #2: No

---

## [Author Response · Author response to Decision Letter 0]

11 Apr 2024

We wish to express our sincere appreciation to the editor and reviewer for their insightful comments. We have done our best to address all the issues raised and we hope that the final version of this manuscript is much improved due to their comments. Attached is the rebuttal letter with responses to all comments

---

## [Decision Letter · Decision Letter 1]

18 Jun 2024

The prevalence of depression, anxiety, and stress and their risks and protective factors among secondary students in Rwanda during the first wave of the COVID-19 pandemic.

PONE-D-23-41465R1

Dear Dr. Niyonsenga,

We’re pleased to inform you that your manuscript has been judged scientifically suitable for publication and will be formally accepted for publication once it meets all outstanding technical requirements.

Kind regards,

Wudneh Simegn, MSc

Academic Editor

PLOS ONE

Additional Editor Comments (optional):

Reviewers' comments:

Reviewer's Responses to Questions

**Comments to the Author**

1. If the authors have adequately addressed your comments raised in a previous round of review and you feel that this manuscript is now acceptable for publication, you may indicate that here to bypass the “Comments to the Author” section, enter your conflict of interest statement in the “Confidential to Editor” section, and submit your "Accept" recommendation.

Reviewer #1: All comments have been addressed

2. Is the manuscript technically sound, and do the data support the conclusions?

Reviewer #1: Yes

3. Has the statistical analysis been performed appropriately and rigorously? 

Reviewer #1: Yes

4. Have the authors made all data underlying the findings in their manuscript fully available?

Reviewer #1: Yes

5. Is the manuscript presented in an intelligible fashion and written in standard English?

Reviewer #1: Yes

6. Review Comments to the Author

Reviewer #1: (No Response)

7. PLOS authors have the option to publish the peer review history of their article (what does this mean?). If published, this will include your full peer review and any attached files.

Reviewer #1: **Yes: **Asmamaw Emagn Kasahun

---

## [Editor Report · Acceptance letter]

21 Jun 2024

PONE-D-23-41465R1 

PLOS ONE

Dear Dr. Niyonsenga, 

I'm pleased to inform you that your manuscript has been deemed suitable for publication in PLOS ONE. Congratulations! Your manuscript is now being handed over to our production team.

Kind regards, 

on behalf of

Dr. Wudneh Simegn 

Academic Editor

PLOS ONE